# On Separate Normalization in Self-supervised Transformers

**Xiaohui Chen**
Department of Computer Science
Tufts University
Medford, MA 02155
xiaohui.chen@tufts.edu

**Yinkai Wang**
Department of Computer Science
Tufts University
Medford, MA 02155
yinkai.wang@tufts.edu

**Yuanqi Du**
Department of Computer Science
Cornell University
Ithaca, NY 14850
yd392@cornell.edu

**Soha Hassoun**
Department of Computer Science
Tufts University
Medford, MA 02155
soha.hassoun@tufts.edu

**Li-Ping Liu**
Department of Computer Science
Tufts University
Medford, MA 02155
liping.liu@tufts.edu

## Abstract

Self-supervised training methods for transformers have demonstrated remarkable performance across various domains. Previous transformer-based models, such as masked autoencoders (MAE), typically utilize a single normalization layer for both the class token [CLS] and the tokens. We propose in this paper a new yet simple normalization method that separately normalizes embedding vectors respectively corresponding to normal tokens and the [CLS] token, in order to better capture their distinct characteristics and enhance downstream task performance. Our empirical study shows that the [CLS] embeddings learned with our separate normalization layer better encode the global contextual information and are distributed more uniformly in its anisotropic space. When the conventional normalization layer is replaced with a separate normalization layer, we observe an average 2.7% performance improvement in learning tasks from the image, natural language, and graph domains.

## 1 Introduction

Transformer models [Vaswani et al., 2017] have revolutionized natural language processing (NLP) [Devlin et al., 2018, Liu et al., 2019] and demonstrated remarkable performances across a wide range of NLP tasks. The significance of transformer models lies in their ability to model context and capture complex linguistic patterns without being constrained by the sequential nature of data. Beyond NLP transformers have further found their successes in areas such as computer vision (CV) [Han et al., 2022], speech recognition [Karmakar et al., 2021], and recommendation systems [Sun et al., 2019, Gu et al., 2020, Wu et al., 2020]. Their flexible architecture and ability to capture dependencies have made them adaptable to diverse data modalities in these domains.

37th Conference on Neural Information Processing Systems (NeurIPS 2023).

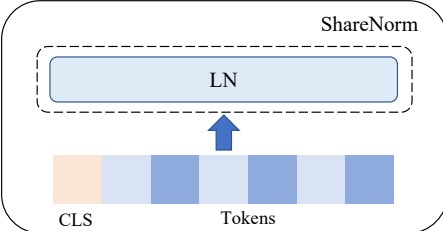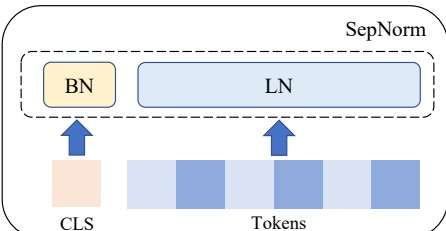

Figure 1: Comparison of the shared normalization (ShareNorm, left) and the proposed separate normalization (SepNorm, right) configurations for token normalization. In the ShareNorm setup, both the [CLS] symbol and other tokens are normalized using a single-layer normalization. In the SepNorm setup, normalization is done separately: the [CLS] symbol is normalized through batch normalization, while other tokens are normalized via layer normalization.

Transformer architectures have been studied extensively from various perspectives such as attention mechanisms, positional encoding [Devlin et al., 2018], and normalization techniques. Specifically, layer normalization [Ba et al., 2016] and batch normalization [Ioffe and Szegedy, 2015] are employed to enhance stability and speed up convergence during training. The literature on transformers also explores parameter initialization [Xu et al., 2019a], optimization algorithms [Huang et al., 2020], regularization techniques [Steiner et al., 2021, Zhou et al., 2020], and improved architectures [Han et al., 2021]. This collective research has advanced transformer architectures and their applications in NLP, CV, and other learning domains.

The study of normalization in transformer architectures is motivated by several factors [Xiong et al., 2020, Shen et al., 2020, Nguyen and Salazar, 2019]. For example, Xiong et al. [2020] emphasize the importance of the warm-up of the learning rate and the position of layer normalization layers for the purpose of stable training and faster convergence. Shen et al. [2020] investigates the disadvantage of using batch normalization in transformers and proposes power normalization. While most previous works focus on how the normalization layer can be modified to stabilize the training process, it is less understood how the normalization affects the encoding abilities of these embeddings.

In self-supervised transformers, the [CLS] symbol is frequently used as a global representation for various downstream tasks [Devlin et al., 2018, He et al., 2022]. Often, the normalization applied to the [CLS] symbol is shared with the rest of the tokens in the sequence, which we term it as Shared Normalization (ShareNorm). Given that the [CLS] symbol plays a special role in representation learning, a natural question is whether we should treat it separately in the normalization operation. Driven by this question, our research first scrutinizes the behavior of the current shared normalization in transformers, particularly the properties of the [CLS] embedding and its influence on downstream task performance. Subsequently, we propose a replacement of ShareNorm with Separate Normalization (SepNorm), the latter of which employs distinct normalization operations for the [CLS] symbol and the token features, as depicted in Figure 1. Through extensive analysis, we demonstrate that [CLS] embeddings learned using ShareNorm have the issue of dimensional collapse, which cannot be rectified even by enforcing uniformity [Wang and Isola, 2020]. However, the straightforward substitution of SepNorm for ShareNorm substantially mitigates this issue. We empirically validate the effectiveness of SepNorm in tasks from the image, text, and graph domains, demonstrating the universal advantage of the proposed SepNorm.

## 2  Background

### 2.1  Pretraining Transformers with the [CLS] symbol

Unsupervised pretraining of a transformer-based model [Vaswani et al., 2017] is widely investigated in many domains, including NLP, computer vision (CV), and graphs.

**Pretraining BERT for NLP.**  In NLP, Devlin et al. [2018] first develop the BERT model by pretraining a transformer-based network by performing the following two tasks – masked language modeling and next sentence prediction. During pretraining, BERT takes a pair of sentences $(\mathbf{x}, \mathbf{y})$,

which are represented as a special sequence

$$\mathbf{s} = \big([\text{CLS}], \mathbf{x}, [\text{SEP}], \mathbf{y}\big). \tag{1}$$

Here [SEP] is a special token that separates the two sentences. A fraction (e.g., 15%) of the tokens in $\mathbf{x}$ and $\mathbf{y}$ are randomly replaced by a special symbol [MASK]. The first task in BERT is to predict the original tokens replaced by [MASK] with cross-entropy loss. The second task is to predict whether $\mathbf{y}$ is the next sentence following $\mathbf{x}$, and the decision is made by classifying the final embedding of the [CLS] symbol. After pretraining, the representation in the [CLS] is usually used for sentence-level downstream tasks such as sentiment analysis [Medhat et al., 2014].

**Pretraining MAE for CV.** The Vision Transformer (ViT) [Dosovitskiy et al., 2020] applies the transformers to computer vision tasks. In ViT, an image is usually voxelized into $16 \times 16$ patches, which are then flattened into a sequence of 256 tokens and fed into the ViT. He et al. [2022] proposes a self-supervised training scheme, Masked Autoencoder (MAE), for the ViT architecture. A training image has 75% of its patches masked. The MAE feeds Tokens of unmasked patches as well as a [CLS] token into the encoder and gets the representations for these tokens. Then the decoder tries to reconstruct the original image by minimizing the mean square error (MSE). Only the encoder will be used for downstream tasks after pretraining. The [CLS] symbol is treated as the class token for linear probing and fine-tuning in the downstream tasks.

**Pretraining Graphormer for molecule discovery.** Graphormer [Ying et al., 2021] is a transformer-based model designed for graph representation learning tasks. It is used to predict the property of a graph rather than a node or edge. Specifically, Graphormer introduces a new symbol [VNode] as a node connecting to all original graph nodes. Then the vector learned for [VNode] represents the global information of the entire graph. The mechanism of [VNode] is similar to the [CLS] symbol in BERT and MAE.

In typical applications of transformers, the [CLS] symbol is not a natural data token. It summarizes other tokens to capture global information, which is especially useful in downstream tasks. For these reasons, we argue that it should be treated differently in normalization operations.

## 2.2 Normalization Layers in Transformers

Given that transformers are initially proposed for NLP tasks, layer normalization (LN) Ba et al. [2016] is typically the normalization method of choice [Xiong et al., 2020]. LN normalizes across feature dimensions and is independent of the sequence length and the batch size. For any features $\mathbf{h} \in \mathbb{R}^d$, the LN has the following computation:

$$\text{LN}(\mathbf{h}) = \boldsymbol{\gamma} \odot \frac{\mathbf{h} - \mu}{\sigma} + \boldsymbol{\beta}, \quad \mu = \frac{1}{d}\sum_{i=1}^{d} h_i, \quad \sigma = \sqrt{\frac{1}{d}\sum_{i=1}^{d}\big(h_i - \mu\big)^2}. \tag{2}$$

Here $h_i$ is the $i$-th dimension of $\mathbf{h}$, $\odot$ represents element-wise multiplication, and $\boldsymbol{\gamma}, \boldsymbol{\beta} \in \mathbb{R}^d$ are scale and bias parameters, respectively. In a transformer, all tokens, including special tokens, such as [CLS] and [SEP], are all treated equally and share the same LNs.

Batch Normalization (BN) [Ioffe and Szegedy, 2015] works by normalizing the input data to have zero mean and unit variance along the batch dimension, followed by an affine transformation to scale the result using gamma and beta parameters. BN normalizes a given vector $\mathbf{h}$ as:

$$\text{BN}(\mathbf{h}) = \boldsymbol{\gamma} \odot \frac{\mathbf{h} - \boldsymbol{\mu}_B}{\boldsymbol{\sigma}_B} + \boldsymbol{\beta}. \tag{3}$$

Here $\boldsymbol{\mu}_B, \boldsymbol{\sigma}_B^2 \in \mathbb{R}^d$ are the running statistics (mean and variance) maintained by the BN. The running mean and variance are updated during training after each batch. They are usually calculated as an exponential moving average of the batch mean and variance. BN is widely adopted in CV but leads to significant performance degradation when naively used in NLP.

## 2.3 Uniformity of the Learned Representations

The dimensional collapse in self-supervised representation learning is a common phenomenon where the embedding vectors only span a lower-dimensional subspace [Jing et al., 2021] of the entire vector

space. This means that the model fails to capture data patterns with full power and instead collapses to a simpler representation. Contrastive methods [Oord et al., 2018, Chen et al., 2020b] have been one of the standard approaches to address this problem. Specifically, Wang and Isola [2020] propose the *uniformity* metric (loss) to quantify the degree of dimensional collapse. Given a set of representation vectors $\{\mathbf{h}_1, \ldots, \mathbf{h}_N\}$ from a dataset of size $N$, the uniformity metric $\mathcal{L}_{\mathcal{U}}$ is computed as follows:

$$\mathcal{L}_{\mathcal{U}} = \log \frac{1}{N(N-1)/2} \sum_{\substack{n=1, \\ m=n+1}}^{N,N} \exp^{-2\left\| \frac{\mathbf{h}_n}{\|\mathbf{h}_n\|} - \frac{\mathbf{h}_m}{\|\mathbf{h}_m\|} \right\|^2}. \tag{4}$$

If the distribution of the representation is perfectly uniform, then the numerical value of $\mathcal{L}_{\mathcal{U}}$ will converge to -4 as the dimension of $\mathbf{h}$ increases to infinity [Wang and Isola, 2020].

In self-supervised transformers, the uniformity of the representation is also taken into consideration by some works. For example, Gao et al. [2021] finetune the pretrained BERT model using the InfoNCE loss [Oord et al., 2018], and Zhang et al. [2022] jointly train the MAE loss along with uniformity loss.

## 3 Approach

### 3.1 Separate Normalization

We present SepNorm, a normalization scheme that separately normalizes embeddings of the [CLS] symbol and embeddings of other tokens. In this work, we focus on the exploration of combinations of BN and LN for the two separate normalization channels.

For instance, if we apply BN to the [CLS] symbol and LN to other tokens, the learnable parameters are structured as $g_1 = (\boldsymbol{\gamma}_1, \boldsymbol{\beta}_1)$ and $g_2 = (\boldsymbol{\gamma}_2, \boldsymbol{\beta}_2)$. Let $\mathbf{H} \in \mathbb{R}^{L \times d}$ represent the feature sequence, where $L$ denotes the sequence length, and $d$ is the feature dimension. Assume embedding $\mathbf{H}_0$ in the first position corresponds to the [CLS] symbol. The normalization process is as follows:

$$\mathbf{H}' = \big(\mathrm{BN}(\mathbf{H}_0; g_1), \mathrm{LN}(\mathbf{H}_1; g_2), \ldots, \mathrm{LN}(\mathbf{H}_L; g_2)\big), \tag{5}$$

where $\mathbf{H}'$ denotes the normalized features. We can also run separate normalization with one of the three other combinations:

$$\mathbf{H}' = \big(\mathrm{BN}(\mathbf{H}_0; g_1), \mathrm{BN}(\mathbf{H}_1; g_2), \ldots, \mathrm{BN}(\mathbf{H}_L; g_2)\big),$$
$$\mathbf{H}' = \big(\mathrm{LN}(\mathbf{H}_0; g_1), \mathrm{BN}(\mathbf{H}_1; g_2), \ldots, \mathrm{BN}(\mathbf{H}_L; g_2)\big),$$
$$\mathbf{H}' = \big(\mathrm{LN}(\mathbf{H}_0; g_1), \mathrm{LN}(\mathbf{H}_1; g_2), \ldots, \mathrm{LN}(\mathbf{H}_L; g_2)\big).$$

Separate normalization allows the [CLS] features to be encoded distinctly from other tokens.

As a comparison, the [CLS] token's embedding and other tokens' embeddings interfere with each other in a shared normalization structure. With ShareNorm, the update directions of the LN parameters $\{\boldsymbol{\gamma}, \boldsymbol{\beta}\}$ are primarily driven by the embeddings of normal tokens. Below is the gradient calculation for these parameters,

$$\frac{\delta\mathcal{L}}{\delta\gamma_i} = \sum_{l=1}^{L} \frac{\delta\mathcal{L}}{\delta\tilde{\mathbf{H}}_{l,i}} \tilde{\mathbf{H}}_{l,i}, \quad \frac{\delta\mathcal{L}}{\delta\beta_i} = \sum_{l=1}^{L} \frac{\delta\mathcal{L}}{\delta\tilde{\mathbf{H}}_{l,i}}, \tag{6}$$

$$\text{where } \tilde{\mathbf{H}}_{l,i} = \frac{\mathbf{H}_{l,i} - \mu_l}{\sigma_l}, \mu_l = \frac{1}{d}\sum_{i=1}^{d} \mathbf{H}_{l,i}, \sigma_l = \sqrt{\frac{1}{d}\sum_{i=1}^{d}\big(\mathbf{H}_{l,i} - \mu_l\big)^2}. \tag{7}$$

We see the summation in the gradient calculation is dominated by normal tokens given that the number of normal tokens is typically a large number. Given the potentially diverse characteristics (i.e., mean and scale) of feature distributions, it might be challenging for normalization parameters to accommodate both token types simultaneously. Moreover, mapping two types of token features into the same sphere may also mix the signal of [CLS] tokens with other tokens. Figure 2(a, b) demonstrates this phenomenon in the scenario where both token types utilize a ShareNorm and how using SepNorm mitigates this effect.

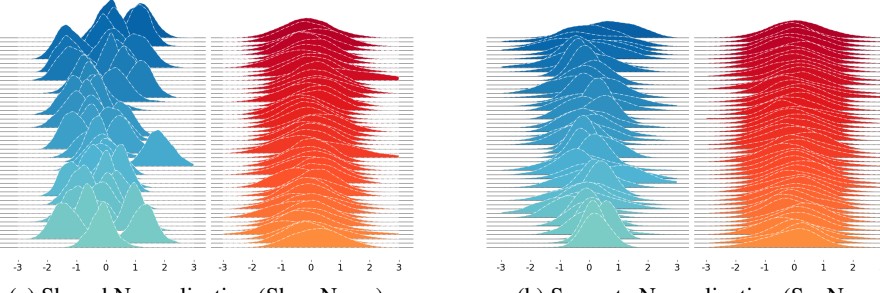

(a) Shared Normalization (ShareNorm)     (b) Separate Normalization (SepNorm)

Figure 2: The effect of SepNorm on feature distributions. Each subplot shows the distributions of the first 50 feature dimensions: [CLS] features are in blue, and other tokens' features are in red. The [CLS] features of ShareNorm are more concentrated around the mean and the mean deviates more from the zero centers, while in SepNorm, the [CLS] distribution is more centered and flattened.

## 3.2 Encourage the Uniformity of the [CLS] Embeddings via a Constrastive Term

We further relate SepNorm with the uniformity of embeddings. Higher uniformity values indicate that embeddings can better exploit the space to store information. Contrastive methods often employ negative instances to encourage uniformity. In particular, we incorporate SepNorm into transformers trained with U-MAE [Zhang et al., 2022], which uses a constrastive term to promote uniformity of features.

The U-MAE explicitly adds a uniformity loss term $\mathcal{L}_{\text{unif}}$ to the training objective to encourage uniformity of [CLS] embeddings.

$$\mathcal{L}_{\text{U-MAE}} = \mathcal{L}_{\text{MAE}} + \lambda \mathcal{L}_{\text{unif}}, \quad \text{with } \mathcal{L}_{\text{unif}} = \mathbb{E}_i \left[ \mathbb{E}_j \left[ \mathbf{h}_{\text{CLS},i}^\top \mathbf{h}_{\text{CLS},j} \right] \right] \tag{8}$$

Here $\mathcal{L}_{\text{MAE}}$ is the MAE training objective. The two indices $i$ and $j$ represent two sequences within the same batch. [CLS] embeddings $\mathbf{h}_{\text{CLS},i}$ and $\mathbf{h}_{\text{CLS},j}$, which are respectively for the two sequences, are obtained from our SepNorm during the transformer calculation. By minimizing $\mathcal{L}_{\text{unif}}$, [CLS] features tend to be different from each other.

## 4 Experiments

We examine the effectiveness of the proposed SepNorm component in three domains: CV, NLP, and graphs. We then further investigate how the ShareNorm and SepNorm affect the uniformity of the [CLS] embeddings.

### 4.1 Computer Vision

**Datasets.** We investigate the model performance on the four image datasets: STL10 [Coates et al., 2011], FGVC Aircraft [Maji et al., 2013], Street View House Numbers (SVHN) [Netzer et al., 2011], and Oxford 102 Flowers [Nilsback and Zisserman, 2008]. All four datasets are for classification tasks. We follow the train/test split provided in the papers introducing the datasets. We report top-1 and top-5 accuracy for all datasets.

**Vision transformers (ViT) and MAE.** We choose Vision Transformer (ViT) [Dosovitskiy et al., 2020] as our feature extractor for all datasets. To pretrain the ViT, we adopt the MAE training scheme [He et al., 2022]. We follow MAE and use a 75% masking ratio on input image. During the downstream tasks, we use the embeddings of the [CLS] token to predict the class labels.

**Experiment setup.** We follow the setup in He et al. [2022] to pretrain and evaluate the ViT. For pertaining, we train the ViT for 4000 epochs. For linear probing, we freeze the encoder's weight and train the last layer on the specific datasets for 2000 epochs. We use a batch size of 512 for pretraining and a batch size of 128 for linear probing.

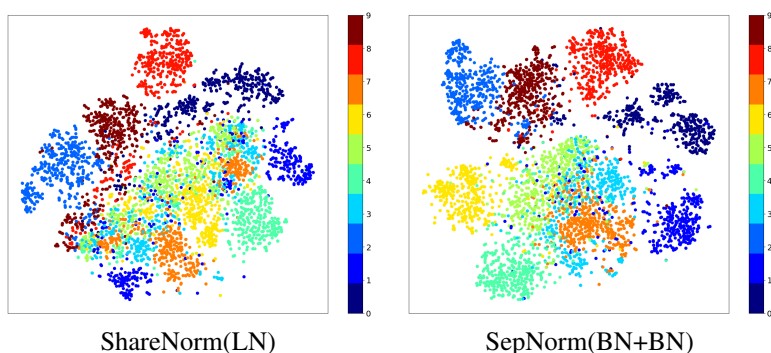

| | ShareNorm(LN) | | SepNorm(BN+BN) | |

Figure 3: t-SNE visualization of representations learned from the STL-10 dataset.

| | STL10 | | Aircraft | | SVHN | | Flower | |
|---|---|---|---|---|---|---|---|---|
| | ACC@1 | ACC@5 | ACC@1 | ACC@5 | ACC@1 | ACC@5 | ACC@1 | ACC@5 |
| MAE | 92.01 | 99.5 | 52.54 | 84.16 | 88.97 | 99.13 | 27.63 | 53.73 |
| + SepNorm | **93.84** | **99.7** | **59.02** | **86.65** | **89.18** | **99.21** | **32.51** | **60.92** |

Table 1: Comparison of linear probing performance of ShareNorm and SepNorm across 4 image classification datasets when the ViT$_{base}$ is pretrained with MAE.

**Experiment results.**  The results presented in Table 1 demonstrate the performances of our model and the baseline model. Our model consistently outperforms the baseline across multiple datasets, indicating its effectiveness in image classification tasks. In the STL-10 dataset, our approach achieves the top-1 accuracy of 93.84% and the top-5 accuracy of 99.7%, higher than the baseline's respective accuracies of 92.01% and 99.5%. Similar improvements are observed in the Aircraft, SVHN, and Flower datasets, where our model consistently outperforms the baseline in both top-1 and top-5 accuracies. These results demonstrate the effectiveness of SepNorm in enhancing image classification performance. We also visualize the embeddings of ShareNorm and SepNorm using t-SNE in Figure 3. Compared with ShareNorm, SepNorm provides embeddings that have better separation among different classes.

## 4.2    Natural Language Processing

**Datasets.**  We evaluated our approach using the STS dataset, which comprises seven semantic textual similarity (STS) tasks. These tasks, including STS 2012-2016 [Agirre et al., 2012, 2013, 2014, 2015, 2016], STS Benchmark [Cer et al., 2017], and SICK-Relatedness [Marelli et al., 2014]. We also evaluate our method on multiple transfer tasks, including MR [Pang and Lee, 2005], CR [Hu and Liu, 2004], SUBJ [Pang and Lee, 2004], MPQA [Wiebe et al., 2005], SST-2 [Socher et al., 2013], TREC Voorhees and Tice [2000], and MRPC [Dolan and Brockett, 2005]. Following the evaluation settings of SimCSE [Gao et al., 2021], we use Spearman's correlation coefficient as the evaluation metric.

**BERT and RoBERTa.**  We conduct our study with pretrained checkpoints of BERT (uncased) [Devlin et al., 2018] and RoBERTa (cased) [Liu et al., 2019], instead of training them from scratch. Using pretrained models is common in this research field [Gao et al., 2021] because the findings are compatible with the common practice of finetuning pretrained models in actual learning tasks. This strategy also saves significant training time and computational resources, allowing us to extend the study to more learning tasks.

**Experiment setup.**  We follow the experiment setup in Gao et al. [2021] and further finetune the BERT and RoBERTa models on English Wikipedia. We evaluate the models using established STS tasks and employ standard evaluation metrics such as Spearman's correlation.

|  |  | STS12 | STS13 | STS14 | STS15 | STS16 | STS-B | SICK-R | Avg. |
|---|---|---|---|---|---|---|---|---|---|
|  |  | | | | Unsupervised Training | | | | |
| BERT$_{base}$ | ShareNorm | 65.28 | 78.82 | 69.65 | 79.02 | 77.21 | 76.4 | **71.74** | 74.04 |
|  | SepNorm | **67.01** | **82.16** | **72.48** | **81.38** | **79.11** | **77.56** | 71.36 | **75.87** |
| RoBERTa$_{base}$ | ShareNorm | **68.25** | 81.24 | 72.78 | 81.38 | **80.31** | 79.83 | 68.16 | 76.00 |
|  | SepNorm | 66.63 | **82.40** | **74.47** | **82.39** | **80.44** | **81.14** | **69.44** | **76.70** |
|  |  | | | | Supervised Training | | | | |
| BERT$_{base}$ | ShareNorm | **77.72** | 81.07 | **78.97** | **85.15** | **82.00** | 82.36 | **79.74** | 81.00 |
|  | SepNorm | 75.32 | **84.41** | 79.94 | 84.91 | 80.87 | **83.63** | 79.61 | **81.23** |
| RoBERTa$_{base}$ | ShareNorm | **77.38** | 80.87 | 78.72 | 84.02 | **82.56** | 83.08 | 78.25 | 80.70 |
|  | SepNorm | 75.80 | **84.94** | **80.33** | **85.51** | 82.11 | **84.88** | **79.72** | **81.90** |
|  |  | MR | CR | SUBJ | MPQA | SST2 | TREC | MRPC | Avg. |
|  |  | | | | Transfer Learning | | | | |
| BERT$_{base}$ | ShareNorm | 82.78 | 88.79 | **94.69** | 89.86 | 87.94 | **84.44** | **75.99** | **86.36** |
|  | SepNorm | **82.82** | **89.08** | 94.30 | 89.70 | **87.97** | 83.88 | 75.21 | 86.14 |
| RoBERTa$_{base}$ | ShareNorm | 84.45 | **91.50** | 93.94 | **89.45** | 90.96 | 86.80 | **76.13** | 87.61 |
|  | SepNorm | **85.11** | **91.56** | 94.30 | 89.43 | **91.66** | **90.96** | 75.58 | **88.37** |

Table 2: Sentence embedding performance on STS tasks and transfer tasks.

**Experiment results.** The experiment results presented in Table 2 highlight the performance of our model compared to the SimCSE baseline on NLP tasks. With the SepNorm layer, BERT$_{base}$ and RoBERTa$_{base}$ achieve overall higher average accuracy compared to ShareNorm's average accuracy. Only in the transfer learning tasks, SepNorm works slightly worse than ShareNorm in BERT$_{base}$, but the difference is marginal.

## 4.3 Prediction of Molecule Properties

**Datasets.** We conducted experiments using the ZINC dataset [Irwin and Shoichet, 2005], which contains approximately 250,000 molecular graphs. The task is to predict the properties of molecules from their graphs. We use a subset of 12,000 molecular graphs, as recommended by the benchmarking methodology outlined in [Dwivedi et al., 2020], so that our results are comparable with other studies. Despite being smaller, the subset retained sufficient diversity and complexity for effective evaluation. We also the MolHIV dataset from the OGB [Hu et al., 2020a] collection, which is widely used for training and evaluating graph-based models in molecular property prediction tasks.

**Graphormer.** We use Graphormer [Ying et al., 2021] as the transformer backbone to construct the predicting model. To obtain graph-level information, Graphormer adds a special node [VNode] to the graph and connects it to all normal graph nodes. The embedding of [VNode] is a summary of the entire graph and will be used in downstream classification tasks. The special node [VNode] serves the same purpose as the [CLS] token in traditional Transformer models. Graphormer has used three encodings to enhance the transformer's learning ability: centrality encoding captures node importance, spatial encoding considers spatial relations, and edge encoding incorporates edge features.

**Experiment setup.** We strictly follow Graphormer [Ying et al., 2021] in terms of the model architecture, hyperparameters, and training strategies. We replaced the ShareNorm in Graphormer with SepNorm to investigate the effectiveness of the proposed component. We evaluate the pretrained model on a broad class of graph-level prediction tasks. We report the mean absolute error for the ZINC and ZINC (subset) datasets and the area under the curve (AUC) for the MolHIV dataset.

| Dataset | ZINC | ZINC (subset) | MolHIV |
|---|---|---|---|
| Metrics | | Mean absolute error↓ | AUC↑ |
| Graphormer | 0.069 | 0.164 | 73.36% |
| + SepNorm | **0.052** | **0.144** | **75.64%** |

Table 3: A comparison of ShareNorm and SepNorm in three tasks of graph property prediction.

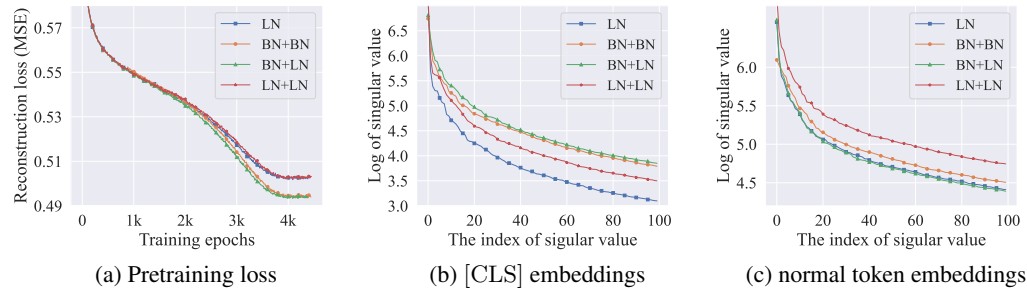

|  | (a) Pretraining loss | (b) [CLS] embeddings | (c) normal token embeddings |

Figure 4: **(a)** Reconstruction loss of the MAE pertaining – MAE with SepNorm achieves lower MSE loss compared to ShareNorm, demonstrating a better ability to encode global contextual information. **(b) & (c)** Comparison of the singular values of learned ([CLS] and normal token) features with ShareNorm and different configurations of SepNorm. [CLS] embeddings learned from SepNorm have larger singular values, which suggests that vectors are better used to encode information.

**Experiment results.** Table 3 shows the performances of our model and the Graphormer baseline. For the ZINC datasets, Graphormer with SepNorm achieves a significantly lower mean absolute error compared to that with ShareNorm. On the MolHIV dataset, SepNorm also improves the AUC to 75.64%, compared with ShareNorm's AUC of 73.36%. These results are strong evidence that the embeddings of the [VNode] can better summarize the properties of the entire graph and thus give superior performance on downstream tasks.

## 4.4 Uniformity Analysis

In this section, we investigate how, under both non-contrastive and contrastive training methods, ShareNorm and SepNorm respectively affect the uniformity of learned embeddings and further classification performances.

**Experiment setup.** We pretrain MAE on the STL10 dataset via four different losses:

- MAE loss $\mathcal{L}_{\text{MAE}}$ without any $\mathcal{L}_{\text{unif}}$ on [CLS] and token embeddings. This setting is a study with MAE training only.
- MAE loss $\mathcal{L}_{\text{MAE}}$ with $\mathcal{L}_{\text{unif}}$ on the [CLS] embeddings. We treat all [CLS] embeddings within the same batch (except itself) as negative instances.
- MAE loss $\mathcal{L}_{\text{MAE}}$ with $\mathcal{L}_{\text{unif}}$ on the token embeddings. We treat all token embeddings within the same batch or same images (except itself) as negative instances.
- MAE loss $\mathcal{L}_{\text{MAE}}$ with $\mathcal{L}_{\text{unif}}$ on both [CLS] and token embeddings.

We choose $\lambda = \{0, 0.01, 0.1, 1\}$. Note that the second loss with $\lambda = 0.1$ corresponds to the U-MAE [Zhang et al., 2022]. We also replace the normalization layer of the ViT in MAE with one of the following: {LN, BN, BN+LN, BN+BN}. The combination of different losses, different $\lambda$'s, and different normalization layers yields 40 specifications of the experiments.

We first report our results with MAE training only. The uniformity of learned embeddings is first measured by singular values of the decomposition of an embedding matrix: we randomly choose 10k embeddings to form the matrix. We do this separately for [CLS] embeddings and normal token embeddings. Figrue 4 shows the results, which indicate that [CLS] features learned from SepNorm exhibit better representational power and thus can better encode the global information.

Then the uniformity is measured by the score in Eqn. 4. Table 6(a) shows the numerical value of the uniformity on the STL10 and Aircraft datasets [Coates et al., 2011, Maji et al., 2013]. Compared to ShareNorm, **SepNorm significantly enhances the uniformity of** [CLS] **embeddings**. Interestingly, the uniformity of normal tokens' embeddings remains comparable. We also empirically verify that better uniformity on the [CLS] embeddings results in better performance on the downstream task (Figure 6(b)). Another observation is that the uniformity of [CLS] embeddings is clearly improved when they are normalized by BN instead of LN. Our hypothesis is that BN tries to make each feature dimension useful by controlling its variance while LN may still neglect some feature dimensions.

| NormLayer | STL10 [CLS] | STL10 token | Aircraft [CLS] | Aircraft token |
|---|---|---|---|---|
| LN | -2.5911 | -3.4998 | -0.9097 | -3.3514 |
| BN | -2.9531 | **-3.6785** | -1.5321 | -3.3659 |
| LN + LN | -1.1045 | -3.3685 | -1.6874 | -3.3633 |
| LN + BN | -1.3153 | -3.3735 | -1.9634 | -3.5056 |
| BN + LN | -3.4426 | -3.3797 | -2.7705 | -3.3068 |
| BN + BN | **-3.4763** | -3.4758 | **-2.9788** | **-3.7349** |

(a) Uniformity of different normalization settings.

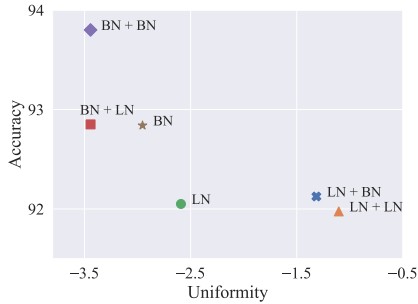

(b) Uniformity vs. accuracy

Figure 6: Uniformity Analysis. **(a)** Under SepNorm, the uniformity of the [CLS] embeddings are better preserved on the STL10 and Aircraft datasets. **(b)** Uniformity is positively related to the downstream task performance – lower uniformity results in higher classification accuracy on the STL10 dataset.

We then report results from studies with U-MAE training. Figure 5 shows the uniformity metrics obtained using different $\lambda$'s. When using ShareNorm, the uniformity of the [CLS] embeddings is no better than -3.088, and even the explicit uniformity loss does not help much. On the contrary, embeddings learned from the proposed SepNorm can easily achieve better uniformity scores. The study with the contrastive approach further verifies the advantage of SepNorm in terms of encouraging uniformity of [CLS] embeddings.

The results provide strong evidence that **the uniformity of the [CLS] embeddings is held down by ShareNorm even the minimization of an explicit contrastive loss cannot increase it**. We

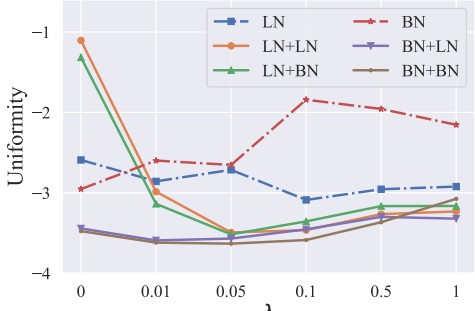

Figure 5: $\lambda$ vs uniformity. A higher $\lambda$ emphasizes more on the uniformity of [CLS] embeddings.

hypothesize that all features after LN will distribute in the same sphere, and [CLS] embeddings are squeezed to a small area of the sphere surface because they need to be different from embeddings of normal tokens.

Table 4 reports the downstream performance (accuracy) on STL10 across 40 different settings. We summarize our observations: (1) In the non-contrastive method MAE, with proper configuration, the performance of SepNorm is superior to that of ShareNorm. (2) In contrastive methods ($\lambda \neq 0$), SepNorms' advantages are further highlighted. For example, when $\lambda = 1$, the performance of SepNorm (BN+LN) is improved by 1.6% compared to the non-contrastive method. The performance gain in SepNorm (BN+BN) is less obvious as the double BNs already impose implicit uniformity loss on both [CLS] and token embeddings.

In contrast to SepNorm, the performance of ShareNorm is less satisfactory when using contrastive methods. We believe it is very challenging to encourage the two types of embeddings to be uniformly distributed in the same sphere and keep them separable at the same time. (3) The uniformity of the token embeddings is also vital for learning a good [CLS] representation, as evident by SepNorm (BN+LN) gaining accuracy with increasing $\lambda$ on the token embeddings. We hypothesize that by enforcing uniformity, the token embeddings are forced to contain less information about others, which encourages the [CLS] embedding to encode as much information as possible. Our empirical study also shows that, when contrastive loss [Oord et al., 2018] is used to encourage the uniformity of [CLS] features in self-supervised transformers, the difference between BN and LN on [CLS] features is not significant anymore.

# 5 Related Works

The training of transformer architectures with self-supervised learning has seen significant advancements in both contrastive and non-contrastive training. Among self-supervised learning methods,

| Normalization layer | $\lambda = 0$ | Negative pairs | $\lambda = 0.01$ | $\lambda = 0.1$ | $\lambda = 1$ | Best |
|---|---|---|---|---|---|---|
| SepNorm (BN+BN) | 93.84 | token | 93.65 | 94.15 | 93.94 | 94.15 |
| | | [CLS] | 93.73 | 93.85 | 93.93 | 93.93 |
| | | [CLS] + token | 93.40 | 94.25 | 94.28 | 94.28 |
| SepNorm (BN+LN) | 92.80 | token | 92.98 | 93.60 | 94.30 | 94.30 |
| | | [CLS] | 92.98 | 93.48 | 93.36 | 93.48 |
| | | [CLS] + token | 92.74 | 93.18 | 94.40 | **94.40** |
| ShareNorm (BN) | 92.84 | token | 92.48 | 93.38 | 92.78 | 93.38 |
| | | [CLS] | 93.10 | 93.33 | 92.93 | 93.33 |
| | | [CLS] + token | 93.41 | 93.46 | 92.99 | 93.46 |
| ShareNorm (LN) | 92.01 | token | 92.61 | 92.74 | 92.14 | 92.74 |
| | | [CLS] | 92.28 | 92.75 | 92.36 | 92.75 |
| | | [CLS] + token | 92.74 | 92.38 | 92.74 | 92.74 |

Table 4: Ablation study of the effect of [CLS] and token uniformity on the downstream tasks with $\lambda$ varied. We report downstream task accuracy for the STL10 dataset.

non-contrastive ones do not rely on negative samples for learning. They have emerged as a powerful approach for training transformer models and demonstrated remarkable successes in various tasks. BERT [Devlin et al., 2018] and RoBERTa [Liu et al., 2019] were proposed in the NLP domain. Additionally, there are some works focus on the specific task, such as speech recognition [Wang et al., 2020], image generation [Chen et al., 2020a], and heterogeneous graph generation [Hu et al., 2020b].

Contrastive methods on the contrary train networks using positive and negative samples that are constructed without manual labeling. They have also been used to train transformer-based architectures. Gao et al. [2021] and Zhang et al. [2022] make significant strides in natural language processing tasks, while Chen et al. [2021] provide valuable insights into the pre-training of transformers. Meanwhile, the potential of contrastive methods in vision transformers has been demonstrated by Caron et al. [2021] and Radford et al. [2021]. These collective efforts underscore the versatility and efficacy of contrastive methods in self-supervised learning of transformers.

Normalization layers, including layer normalization and batch normalization, are essential to transformer architectures because they help stabilize the training procedure and accelerate convergence. Xiong et al. [2020] delve into the role of layer normalization in the transformer architecture and provide insights about how the layer improves the training stability and the performance of transformers. Similarly, Xu et al. [2019b] explores the intricacies of layer normalization and offers potential enhancements to its effectiveness. To address the limitations of traditional batch normalization in a transformer architecture, Shen et al. [2020] introduces a new normalization layer, Powernorm, which is a variant of batch normalization. Nguyen and Salazar [2019] focus on the normalization process in the self-attention mechanism of transformers and propose methods to optimize the normalization of self-attention. All the efforts above underscore the critical role of normalization layers in transformer models.

## 6 Conclusion

In this work, we have introduced SepNorm to separate the normalization of [CLS] embeddings from that of other tokens. Across three application domains (images, text, and graphs), SepNorm shows consistent performance improvement when it is incorporated into transformer models. Our analysis shows that SepNorm promotes uniformity of [CLS] embeddings and thus enhances the transformers' ability to encode information. As a valuable technique for improving the foundational transformer architecture, SepNorm has the potential to benefit a wide range of applications.

## Acknowledgement

We thank all reviewers for their constructive feedback. This research is supported by the NIGMS of the National Institutes of Health, Awards R35GM148219, the Army Research Office, MURI program, contract # W911NF2210239, and NSF Award 1909536. Chen and Liu are also supported by the NSF CAREER Award # 2239869.

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
