# OpenReview forum: "On Separate Normalization in Self-supervised Transformers"
_NeurIPS.cc/2023/Conference — NeurIPS 2023 poster_

### Official Review · Reviewer_4DDo · 2023-07-03

**Soundness:** 4 excellent
**Presentation:** 3 good
**Contribution:** 3 good
**Rating:** 7
**Confidence:** 4

**Summary:**

The paper proposes a simple modification to Transformer-based self-supervised learning by having distinct normalization (parameters) for the CLS token, which captures the global information for use in downstream tasks, and the remaining tokens. The motivation is that this allows the tokens to avoid dimension collapse as indicated by a lower uniformity score, which in turn correlates with higher downstream performance. The paper verifies in a case study that this is empirically the case.
The proposed method, SepNorm, is evaluated in a controlled setting against the standard approach, ShareNorm, on various datasets from different domains. SepNorm consistently achieves improvements. An additional ablation study confirms that this is also the case when uniformity is encouraged explicitly through an additional loss term.

**Strengths:**

* The paper proposes an extremely simple, yet consistently effective solution to mitigate the problem of dimensional collapse and thereby improve performance.
* The paper is concerned with general self-supervised Transformer architectures, which are relevant to the majority of the NeurIPS audience.
* The experiments are very clear and well executed, the controlled experimental setup leaves little room for doubt regarding the method's merits.
* The paper is generally self-contained, all necessary pieces can be understood by a general ML-savvy audience.
* The idea to use separate sets of learnable parameters across different components of the model is of course not new per se, but the application context and the resulting insights constitute sufficient novelty.

**Weaknesses:**

The only main weakness is that the paper could be structured better. Sections 3.1. and 3.2 mix the description of the proposed method with experimental results that are supposed to motivate the method. However, those results already rely on the existence of the proposed method, and do thus not suit as motivation. Rather, they constitute useful analyses of the proposed method, which could just as well be discussed in the "Experiments" section. This has the added benefit that the datasets used for generating results in Figures 3 and 4 have already been introduced (which only happens in 5.1). In fact, section 3.1. and 3.2. never explicitly mention that these are computer vision datasets.

**Questions:**

* In section 4, alternative normalization methods are discussed. In how far do these address the dimensional collapse problem? How would your results change with different normalizations, i.e., in how far is your method orthogonal to these?
* Table 2: It is unclear what you mean by BERT-base and RoBERTa-base. Do these refer to the original models, or are these finetuned in the same way as the "+SepNorm" models? If it's the former, how do we know that the improvement isn't merely a result of continued finetuning?

**Limitations:**

The paper does not address limitations. I think it would be useful to discuss whether the results would extend to other types of normalization (see question above), and what potential avenues for future work are.

---

> ### Author Rebuttal · Authors · 2023-08-09
>
> We sincerely appreciate the reviewer's positive feedback on our paper, as well as the suggestions that can improve our submission's quality. Below we try our best to address those concerns and questions from the reviewer.
>
> ## [Weakness]
> **Q1: The only main weakness is that the paper could be structured better. Sections 3.1. and 3.2 mix the description of the proposed method with experimental results that are supposed to motivate the method. However, those results already rely on the existence of the proposed method, and do thus not suit as motivation. Rather, they constitute useful analyses of the proposed method, which could just as well be discussed in the "Experiments" section. This has the added benefit that the datasets used for generating results in Figures 3 and 4 have already been introduced (which only happens in 5.1). In fact, section 3.1. and 3.2. never explicitly mention that these are computer vision datasets.**
>
> R1:Thank you for your feedback! We will restructure the methodology section as suggested to enhance the clarity and coherence of our paper. We believe your suggestion will improve the readability of our work.
>
>
> ## [Questions]
> **Q1: In section 4, alternative normalization methods are discussed. In how far do these address the dimensional collapse problem? How would your results change with different normalizations, i.e., in how far is your method orthogonal to these?**
>
> R1: Thank you for your insightful question. Our method is entirely distinct from previous normalization approaches in terms of both its motivation and methodology. Furthermore, our method can be seamlessly integrated with any normalization approach. As far as we know, the alternative normalization methods discussed in Section 4 do not specifically aim to tackle the dimensional collapse problem; instead, they primarily focus on stabilizing the training process of transformers.
>
> To better understand how our method can complement alternative normalization techniques, consider the following insights:
>
> 1. **For non-contrastive self-supervised training**, certain normalization methods like BatchNorm and PowerNorm can also promote uniformity. If PowerNorm demonstrates better training process stabilization than BatchNorm, we recommend using PowerNorm.
> 2. **For contrastive self-supervised training**, it’s empirically shown that LayerNorm may perform better than BatchNorm [1] as it does not utilize batch statistics. This observation may also extend to PowerNorm.
>
> It is worth noting that we have not discussed other normalization methods such as GroupNorm and PairNorm since they are not explicitly designed for transformers.
>
> We hope that this response addresses the reviewer's concerns. Please feel free to inquire about any other questions you may have.
>
>
> **Q2: Table 2: It is unclear what you mean by BERT-base and RoBERTa-base. Do these refer to the original models, or are these finetuned in the same way as the "+SepNorm" models? If it's the former, how do we know that the improvement isn't merely a result of continued finetuning?**
>
> R2: We apologize for the confusion, “+SepNorm" denotes BERT-base with SepNorm, and BERT-base denotes BERT-base with ShareNorm. Both models are finetuned in the same way to make a fair comparison. We have corrected the table in the attached PDF, all the results will be included in the revised script.
>
> ## [Limitations]
> **Q1: The paper does not address limitations. I think it would be useful to discuss whether the results would extend to other types of normalization (see question above), and what potential avenues for future work are.**
>
> R1: We thank the reviewer for the insightful questions. A common limitation of such innovations of network layers is the increased complexities of model tuning: there are more hyperparameters to set. However, we hope the extensive benefit shown in this paper will help the community to settle for a default choice of SepNorm. We will add a paragraph discussing the limitation of the proposed method.
>
> In terms of extending to other types of normalization, we kindly refer the reviewer to [Questions]-Q1 for the discussion of pairing with other types of normalization.
>
> Regarding potential future works, we provide two possible avenues in the following:
>
> 1. Relationship between [CLS] and non-[CLS] tokens, and potential architecture improvement:
> A promising area for future work is exploring the relationship between the [CLS] token and non-[CLS] tokens. Our ablation study indicated that achieving better uniformity in the non-[CLS] tokens could positively impact downstream performance. Additionally, it remains unknown whether dimensional collapse issues exist in token-level tasks such as generation, object detection, etc. If there are any, we suggest investigating whether improvements to transformer designs can address these issues and further enhance performance.
>
> 2. Extension to decoder-only large language models:
> While our current work focuses on encoder-only transformers, we recognize the growing popularity of large language models utilizing decoder-only transformers. To gain a comprehensive understanding of representation distributions and the role of uniformity in generative tasks, it is essential to extend our analysis to these decoder-only architectures. Exploring how uniformity influences the performance of generative tasks in decoder-only transformers may also provide valuable insights into their behavior.
>
> We will revise our manuscript to incorporate those discussions suggested by the reviewer. Thank you for the valuable comments.

---

> > ### Comment · Reviewer_4DDo · 2023-08-15
> >
> > Thank you for discussing limitations and showing the will to improve the readability of the paper. Unfortunately, since the discussion period doesn't allow to update the draft, I can not improve my score based on promises. However, I think the work is fine either way, so I'll keep the score.

---

### Official Review · Reviewer_hvnd · 2023-07-05

**Soundness:** 3 good
**Presentation:** 3 good
**Contribution:** 3 good
**Rating:** 7
**Confidence:** 4

**Summary:**

The paper introduces SepNorm, a normalization technique for Transformer models. SepNorm separates the normalization of the [CLS] token from the rest of the tokens in a sequence, a departure from the traditional ShareNorm method that normalizes all tokens together.

**Strengths:**

1. By applying SepNorm, the embeddings of the [CLS] symbol effectively captures the characteristics of the entire graph, leading to improved results in downstream applications.
2. SepNorm achieves better uniformity by encouraging uniformity for both the [CLS] and token embeddings when used in contrastive methods.
3. SepNorm seems to do well in the evaluation.

**Weaknesses:**

No uncertainty/confidence/error bars on experimental results or significance testing.

**Questions:**

Would SepNorm be helpful for multilingual machine translation tasks?

**Limitations:**

No visualization of the learned representation from before and after applying SepNorm

---

> ### Author Rebuttal · Authors · 2023-08-09
>
> Thank you for the suggestion on our paper, below we address your concern accordingly.
>
> ## [weakness]
> **Q1: No uncertainty/confidence/error bars on experimental results or significance testing.**
>
> R1: As suggested, we re-run the experiments to obtain the standard derivations in the NLP tasks as it is faster to train compared to the tasks in CV and graph domains. Moreover, we've also included the experiment results from the supervised training setting. We kindly ask the reviewer to check the results in the attached PDF.
>
> ## [Questions]
> **Q1: Would SepNorm be helpful for multilingual machine translation tasks?**
>
> R1: Thank you for the insightful question! SepNorm can be better used in discriminative models rather than generative models. BERT originally proposes [CLS] token and is usually used for non-autoregressive transformers (the encoder part in [1]). We didn’t further investigate its use case in decoder-based transformers. Since machine translation usually requires an encoder-decoder model to perform conversion between two languages, it may not be an ideal scenario for applying SepNorm. However, SepNorm can be used for multilingual-related tasks such as alignment [2]. We will further discuss the suitable scenarios for using SepNorm in the future revised manuscript.
>
>
> ## [Limitations]
> **Q1: No visualization of the learned representation from before and after applying SepNorm.**
>
> R1: Thank you for your feedback. In response, we’ve visualized the learned representation of the STL-10 dataset for both ShareNorm and SepNorm. We use t-SNE to reduce the dimension from 784 to 2 before visualization. Class label of each data point are highlighted with different colors. We kindly refer the viewer to the attached PDF for the visualization result.
>
> ## [References]
>
> [1] Vaswani, Ashish, et al. Attention is all you need. NIPS 2017
>
> [2] Cao, Steven, et al. Multilingual alignment of contextual word representations. ICLR 2020

---

> > ### Comment · Reviewer_hvnd · 2023-08-14
> > **Comments after rebuttal**
> >
> > Thank you for your response, which included re-running the experiments to obtain the standard derivations in the NLP tasks and the visualization of the learned representation before and after applying SepNorm. This shows the robustness of SepNorm. For this reason, I am willing to revise my score up.

---

### Official Review · Reviewer_vuG3 · 2023-07-06

**Soundness:** 3 good
**Presentation:** 3 good
**Contribution:** 2 fair
**Rating:** 6
**Confidence:** 4

**Summary:**

The authors propose to use a different normaliser to the [CLS] token and the rest of the tokens for masked autoencoders.
They motivate this as an improvement on the standard normalisation and combine it with contrastive uniformity loss.
They observe some improvements in classification tasks .

**Strengths:**

The pape is proposing a new normalisation of MAE using 2 separated normalisers one for [CLS] token and another for other tokens.
They provide evidence this helps the models to obtain better perfomance indepndently and in combination with uniformity loss.

**Weaknesses:**

The contribution while important doesn't seem relevant for a larger audience but they do per discussion below for several tasks.

**Questions:**

While the [CLS] token stands among others,  it looks like this can be an extreme of different normalisation depending on the frequency.
Have authors compared how much of the improvement comes from the token frequency vs how much comes from the semantic full sentence encoding of the token ?

**Limitations:**

The authors have studied some ablation studies.

---

> ### Author Rebuttal · Authors · 2023-08-09
>
> Thank you for your effort in reviewer our submission. Below we answer the concerns and questions that you raised, please feel free to ask if  you have further questions.
>
> ## [weakness]
> **Q1: The contribution while important doesn't seem relevant for a larger audience.**
>
> R1: Thank you for your feedback. We feel that our proposed method can actually benefit a broad class of audiences in the machine learning community. Our proposed plug-and-play component, SepNorm, has the potential to benefit diverse domains. Our experiment demonstrated that it can be easily applied to various transformer-based architectures (e.g., MAE, BERT, Graphormer) used in different domains (CV, NLP, Graph). By incorporating SepNorm, researchers, and practitioners can improve the performance of these architectures without extensive modifications. We will enhance the clarity and presentation of our findings in the revised version of the paper.
>
> ## [questions]
> **Q1: While the [CLS] token stands among others, it looks like this can be an extreme of different normalisation depending on the frequency. Have authors compared how much of the improvement comes from the token frequency vs how much comes from the semantic full sentence encoding of the token?**
>
> R1: Regarding the "frequency" aspect, are you referring to the occurrence frequency of individual words within the corpus? To rephrase your question, are you asking whether it's necessary to normalize the most frequently appearing word separately or if the special token requires its own normalization? Our response is that the unique [CLS] token necessitates distinctive normalization for two primary reasons. First, the vector representation of the [CLS] token is directly employed in downstream tasks. Second, this vector effectively encodes the comprehensive sentence semantics, desiring a distinct treatment. We are open to providing further clarification if our interpretation of your question is not accurate.

---

> > ### Comment · Reviewer_vuG3 · 2023-08-12
> >
> > Thanks for the additional experiments, the nice work and the time devoted to clarifying the points of concerns.
> >
> > ## [weakness]
> >
> > ### Q1: The contribution while important doesn't seem relevant for a larger audience.
> > ### R1: Thank you for your feedback. We feel that our proposed ...
> >
> > *Reviewer's Answer*:
> >
> > Apologies. I think, in hindsight that I should have added more information into that comment.  I didn't mean the practical implications of this technique will not be interesting or relevant but rather that  the paper may -- from one's person point of view which could potentially be biased -- not be relevant for a large audience for long time w/o follow-ups. The other comments on the review attempt to show glimpse of why. I hope my view is clarified and of course the results that with your hard work and diligence have been collected are surely relevant and interesting for part of the community.  In case of doubt, please note that this has not affected the overall rating, for which I focused on the content itself, given the subjective and risky nature of that comment.
> >
> > ## [questions]
> > ### Q1: While the [CLS] ...
> > ### R1: Regarding the "frequency" aspect,  ....
> > *Reviewer's Answer:*
> >
> > The main question can  be restated as which are the causes and not the effects of the SepNorm approach ? the analysis seems to be focused on uniformity. Given the heuristic nature of the proposed solution and the Uniformity effect analysis, I think other heuristic solutions comparison based on statistical analysis might be beneficial.  Note that because Normalisation layers essentially standardise the input, the improvement can come from a purely statistical effect of taking out statistical outliers, or recursive mean computation, ... .  For instance, one could simply split the tokens into top-k frequent events and less frequent events which might show different distribution behaviour. Since the token [CLS]/[VNode] is a single token and occurs at each training example, it can be biasing the distribution statistically because of the frequency. Alternatively, if the [CLS] activation is the average of other activations, then the Normalization layer might not be able to adapt during training to those dynamics, where x_0 = mean(x1...xN). Note as well that if the sequence lengths are in average 128 and we have N training examples, then there is a strong bias against the [CLS] token activation vs other frequent tokens such as "th" or "a" --among others--; or even 1/128 if all other tokens do share activation distribution except for [CLS]. Consequently, one could scale the contribution of [CLS] proportionally. Furthermore, we could erase the frequency from the update by making parameters depend on the frequency, ... etc ...    These are some of the thoughts, I was considering while writing the main question. Essentially, the SepNorm is simply one solution to the approach that is subsumed in other options when trying to find the causes/solutions for the [CLS] behaviour. Note I have intentionally dropped [SEP] from discussion about. BTW, What about the [SEP] token ? would it be beneficial to split it out as well ?
> >
> > Thanks !!

---

> > > ### Author Response · Authors · 2023-08-14
> > > **Thank you for your comment!**
> > >
> > > ### **Reviewer's explanation on the contribution comment**
> > > #### *Author response*:
> > > Thank you for the response, and we appreciate that you recognize our hard work. We agree that our proposed reasoning may not be unique. However, we would like to emphasize the importance and relevance of this problem (self-supervised transformers, which have been the de facto choice of different research communities for learning from unlabeled data, as demonstrated by our experiments in NLP, CV, and Graph). We also hope that by observing and interpreting the problems, more people would notice this phenomenon and (potentially) other related phenomena in self-supervised transformers. We believe our observation and explanation are self-contained, and we propose a simple yet effective approach to mitigate the issue and demonstrate the empirical performance gains across various tasks. We will add more discussions in the future work section to point out potential extensions and follow-ups of our work.
> > >
> > > ### **Reviewer's clarification on the frequency question**
> > > #### *Author response*:
> > > Thank you for your clarification!
> > >
> > > **Here is our short response**: We advocate treating [CLS]/[SEP] as different variables for other natural tokens.(by natural, we mean the token is originally in the dataset and not introduced heuristically, such as [CLS],[SEP]). But we do not think treating natural tokens differently based on their behavior (for instance, their frequency) is a good idea. While SepNorm works well in promoting representation quality, we agree that it is not the only solution separating the two kinds of variables, and other promising solutions may exist. We’d like to leave such work in the future. We think that apart from SepNorm, another contribution of our work is the finding that the two types of tokens should not share the same embedding space (through uniformity analysis). We hope the reviewer could agree on this.
> > >
> > > Below we further elaborate on the reviewer's comments:
> > >
> > > **(We advocate treating [CLS]/[SEP] as different variables (outliers) against other natural tokens.)**\
> > > Your observation that “the improvement can come from a purely statistical effect of taking out statistical outliers, or recursive mean computation” is insightful. Your assumption that the [CLS] embeddings (outliers) do not belong to the intrinsic distribution of the other token embeddings aligns with our motivation. Our approach takes a further step and assumes those “outliers” belong to another distribution and access the uniformity of the “outliers” distribution.
> > >
> > > **(treating natural tokens differently (as outliers) based on their behavior (for instance, their frequency) & biased fitting)**\
> > > However, we do not feel it’s reasonable to identify “outliers” based on their frequency. Since stopwords like “the” and “a” are frequently observed in the data. Statistically, their high frequency will not bias the fitting of the normalizing parameter because such behavior is the nature of the data. On the contrary, adopting different treatments to those high-frequency may result in biased fitting. Such an argument does not apply to the [CLS]/[SEP] tokens! Since they are heuristics and not originally from the data. We further provide an ablation study that performs SepNorm on stopwords (collected from NLTK) such as “a”, “the”, and “are” tokens. The performance gain is marginal (74.04&rarr;74.18 on unsup. STS with BERT model).
> > >
> > > **(Simple treatment that reweights contribution between [CLS] and other tokens)**\
> > > The remark made by the reviewer that “one could scale the contribution of [CLS] proportionally (when updating the normalization statistics) to improve model performance” is a possible approach. Intuitively, such an approach can determine “the 'volume' the [CLS] token and the other tokens should occupy” by tunning the contribution coefficient. To do this, we increase the weight of the [CLS] token by L/2L (L=256) when optimizing the normalization statistics. Such an operation leads to a performance gain (see table below), but not as good as SepNorm. Moreover, further increasing the important weight might result in performance drops. We believe such an operation requires careful parameter selection to balance the two types of tokens. SepNorm directly separates them into two embedding spaces, ensuring they won’t “compete” for space.
> > >
> > > | | ShareNorm | ShareNorm(L)| ShareNorm(2L)|SepNorm|
> > > |-|:-:|:-:|:-:|:-:|
> > > |**Accuracy**| 92.01 |92.75| 91.88| 93.84|
> > >
> > > **(SepNorm on [SEP] token)**\
> > > In response to the last comment about the [SEP] token, there is no harm in introducing SepNorm to it. *We believe such an operation may help promote the uniformity of the natural token embedding but the downstream improvement gain might be marginal*. Because [CLS] token may benefit less from it since it already has its normalization layer.
> > >
> > > We thank you again for the clarification, and we hope we've answered all the questions. Please do not hesitate to ask if there are more questions.

---

### Official Review · Reviewer_QRJ8 · 2023-07-07

**Soundness:** 2 fair
**Presentation:** 2 fair
**Contribution:** 2 fair
**Rating:** 4
**Confidence:** 4

**Summary:**

The paper proposes a new method called SepNorm, which utilizes separate normalization layers for the [CLS] token and the remaining tokens, replacing the conventional single normalization layer (ShareNorm) in transformers. Experiments demonstrate the importance of applying separate normalization to the [CLS] token and the remaining tokens, and show that SepNorm enhances the transformer's ability to encode the input.

**Strengths:**

The paper propose to explore separate normalization for the [CLS] token in pretrained transformers. Experiments on three different domains  show the effectiveness of the proposed method.

**Weaknesses:**

The paper mentions that "Our method aims to alleviate the potential negative effects of using the same normalization statistics for both token types, which may not be optimally aligned with their individual roles." It seems that this statement is not empirically validated in the paper.

In the ablation study, only experiments on a CV dataset have been conducted. It would be better if more experiments on NLP and Molecule Discovery datasets could be carried out.

The paper only selects one or two models for each domain in the experiments, such as evaluating the STS task in the NLP domain. However, this limited selection of tasks is not sufficient to demonstrate the generalizability of SepNorm. It would be more appropriate to include a wider range of experiments across multiple tasks to provide a comprehensive evaluation of the effectiveness and generalizability of the proposed SepNorm method.

The paper does not empirically compare the proposed method against previous works, such as Powernorm mentioned in the section of related work.



**Questions:**

1. Can the proposed method be scaled to large language models?

2. How is the speed of SepNorm in comparison to ShareNorm?


**Limitations:**

The paper does not discuss the limitations of the proposed method. It would be better to discuss the potential impact of the proposed normalization method on learned representations, training time in addition to the performance on the downstream tasks.

---

> ### Author Rebuttal · Authors · 2023-08-09
>
> ## [weakness]
> **Q1: the statment "Our method aims to alleviate the potential negative effects of using the same normalization statistics for both token types, which may not be optimally aligned with their individual roles." is not empirically validated in the paper.**
>
> R1: The potential negative effect we refer to is **the relatively low uniformity score achieved by using ShareNorm**. Uniformity has been widely used to assess the degree of dimension collapse and is often considered an essential metric for learned representation quality [1,2,3]. We quantitatively showed that the uniformity for ShareNorm is consistently worse than SepNorm, regardless of the emphasis that is placed on the metric. We will modify our statement accordingly to avoid such confusion.
>
> **Q2: It would be better if more experiments on NLP and Molecule Discovery datasets could be carried out.**
>
> R2: Thank you for your comment. As you suggested, we expanded our experiments to cover a broader range of NLP datasets. Due to time constraints, we are not able to explore on Molecule tasks. We will consider incorporating more Molecule datasets in the future. Below, we brief how we further validate our proposed method in various NLP datasets/tasks.
> * Models: We use Bert and RoBERTa models.
> * Input variants:
> We finetune BERT and RoBERTa on multiple datasets using three input settings, using either the [CLS] or [MASK] tokens for sentence-level tasks. For [MASK] predictions, an actual word is predicted, and its semantic meaning determines the label, e.g., "terrible" indicates a negative label.
>   1. Standard: Input is "[CLS] <Sentence> [EOS]", we predict labels via [CLS] embeddings.
>   2. Prompt-based: Input is "[CLS] <Sentence>. This is [MASK] [EOS]", we use [MASK] embeddings for predictions.
>   3. Prompt-based with demo: Input is "[CLS] <Sentence1>. This is [MASK]. [SEP] <demo1>. This is <label1>. [SEP] <demo2>. This is <label2> [EOS]", we use [MASK] embeddings for predictions. Demos are drawn from training data.
> * Datasets:
> We use the following datasets: Stanford Sentiment Treebank 2&5 (SST-2 & SST-5), Movie Reviews(MR), Customer Reviews(CR), Multi-Perspective Question Answering(MPQA), Subjectivity Dataset(SUBJ), Corpus of Linguistic Acceptability(CoLA), Text REtrieval Conference(TREC), Stanford Natural Language Inference(SNLI), Question Natural Language Inference(QNLI), Microsoft Research Paraphrase Corpus(MRPC), Quora Question Pairs(QQP).
> * Results: We report the result in the uploaded pdf in the general response. As we can observe, applying SepNorm on [CLS] can help improve the model performance with different prompting methods.
>
> We hope the additional experiment can address the reviewer’s concern about our method’s effectiveness.
>
> **Q3: Include a wider range of experiments across multiple tasks to comprehensively evaluate the effectiveness and generalizability of SepNorm.**
>
> R3: Thank you for the feedback. In response, we've extended our evaluation to multiple transfer tasks, including MR, CR, SUBJ, MPQA, SST-2, TREC, and MRPC. To do this, we freeze the model weights learned from SimCSE and only train the classifier on top of the model. We kindly refer the reviewer to the attached pdf for more details on the experiment results.
>
> **Q4: The paper does not empirically compare the proposed method against previous works, such as Powernorm...**
>
> R4: We’d like to point out that we do not propose a new normalization layer. Instead, we propose a normalization strategy that can be combined with different normalizations such as BatchNorm, LayerNorm, and PowerNorm. Since there are many combinations of different normalization layers in SepNorm, we simplify the setting and focus on the canonical case.
> ## [questions]
>
> **Q1: Can the proposed method be scaled to large language models?**
>
> R1: Our method can be applied to large language models that utilize the [CLS] tokens. For example, CLIP aligns the [CLS] embeddings from image and text via contrastive learning. SepNorm can be incorporated into the text and vision transformers.
>
> **Q2: How is the speed of SepNorm in comparison to ShareNorm?**
>
> R2: With SepNorm, the model needs no more computation but only stores separate mean and variances with a little extra space. In practice, the computation cost of replacing ShareNorm with SepNorm is negligible. For example, for training an MAE on STL-10 dataset in the machine with one A100 GPU with 32 CPU cores, both normalization methods took 36 hours to finish.
> ## [limitations]
> **Q1: It would be better to discuss the potential impact of the proposed normalization method on learned representations, training time in addition to the performance on the downstream tasks.**
>
> R1: Thank you for the feedback.
> 1. Learned Representations: We quantitatively evaluated representation uniformity, noting that better uniformity indicates better data information preservation[1]. Our qualitative analysis in Figure 2(a,b) shows that SepNorm results in a more uniform [CLS] feature distribution around a mean of 0. Moreover, the slower singular value decay in Figure 2(d,c), suggests that SepNorm can better span the feature space, leading to better representation.
> 2. Training time: The replacement of SepNorm does NOT incur extra computation cost. The extra computation here is to perform another normalization operation over the feature, which is affordable compared to the self-attention modules.
> 3. Performance on the downstream tasks: We verify SepNorm on downstream tasks from three domains. For example, we pretrain the transformers with ShareNorm/SepNorm in the CV and graph domains and finetune them on the corresponding downstream tasks (results are reported in Table 1, 3 in Section 5).
> ## [Reference]
> [1]Wang, et al. Understanding contrastive representation learning through alignment and uniformity on the hypersphere. ICML 2020.
>
> [2]Bojanowski, et al. Unsupervised learning by predicting noise. ICML 2017.
>
> [3]Mettes, et al. Hyperspherical prototype networks. NIPS 2019.

---

> > ### Comment · Reviewer_QRJ8 · 2023-08-16
> > **Thanks for providing new experiment results**
> >
> > The provided new results partially address my concerns. For this I'd like to increase my score. But some results are mixed and the method is quite limited to the scenario with [CLS] token.

---

> > > ### Author Response · Authors · 2023-08-16
> > >
> > > Thank you for your kind response. We are willing and are happy to provide more results if you have other comments on the our additional experiment. Regarding the comment that our proposed method has limited application, we believe the use of [CLS] token is quite universal, which could benefit various domains just as illustrated as in the experiment.

---

### Official Review · Reviewer_voLG · 2023-07-13

**Soundness:** 4 excellent
**Presentation:** 4 excellent
**Contribution:** 4 excellent
**Rating:** 7
**Confidence:** 4

**Summary:**

This paper analyzes the normalization layer that is applied to both the context tokens and the [cls] token in an input. It argues that traditional normalzation that is applied to both type of tokens, dubbed ShareNorm, is not effective since they have different roles. The paper proposes to use separate normalization layers for these tokens, SepNorm, and conducted a series of analysis. Echoing prior work, they showed that a measure of uniformity of the representations correlates with downstream classification performance, and further that ShareNorm find hard to tackle this uniformity deficiency even with uniformity fixing methods, while SepNorm improves uniformity and hence downstream task performances.

**Strengths:**

1. The paper is well written and easy to understand. It starts with sufficient background and explains the analysis clear.
2. The hypothesis intuitively makes sense to me, and the empirical analysis provides good support.
3. Experiments are conducted on a number of tasks in vision, nlp and graphs.

**Weaknesses:**

I don't have major concerns on this work.

**Questions:**

n/a

**Limitations:**

no limitations were discussed.

---

> ### Author Rebuttal · Authors · 2023-08-09
>
> We are sincerely grateful for your time and effort in reviewing our submission. We greatly appreacited your positive feedback on our proposed method, and we are glad to know that you found the paper well-written, easy to understand. Once again, thank you for your kind words and for accepting the paper.

---

> > ### Comment · Reviewer_voLG · 2023-08-16
> >
> > Thanks!

---

### Author Rebuttal · Authors · 2023-08-09

We thank all reviewers for their effort in reviewing our submission. And we appreciate all the positive and negative feedbacks on our manuscript. We've tried our best to address all the concerns and questions raised by the reivewers. The attached pdf includes the additional experiments and visualization asked by reviewer QRJ8 and hvnd. Thanks again for the valuable feedbacks.

---

### Author Response · Authors · 2023-08-21
**Thanks for the valuable comments and fruitful discussions**

Dear Reviewers,

We wish to take this opportunity to thank the reviewer for their extensive constructive feedback during the review process, for raising interesting and relevant questions for the work, and for highlighting a number of action points which have served to strengthen the paper.

Best,

Authors

---

### Decision · Program_Chairs · 2023-09-21

**Decision:**

Accept (poster)

**Comment:**

The paper proposes a simple and straightforward modification to Transformer-based self-supervised learning by having distinct normalization (parameters) for the CLS token, which captures the global information for use in downstream tasks, and the remaining tokens. The proposed method, SepNorm, is evaluated in a controlled setting against the standard approach, ShareNorm, on various datasets from different domains.

After the response, the authors added more experiments on NLP tasks, which addressed the concerns of reviewers.